
**Measurement report: Long-term measurements of aerosol precursor concentrations in the Finnish sub-**
**Arctic boreal forest**
Tuija Jokinen[1,2*], Katrianne Lehtipalo[1,3], Roseline Cutting Thakur[1], Ilona Ylivinkka[1], Kimmo Neitola[1], Nina
Sarnela[1], Totti Laitinen[1], Markku Kulmala[1], Tuukka Petäjä[1] and Mikko Sipilä[1]
[1]Institute for Atmospheric and Earth System Research (INAR) / Physics, Faculty of Science, University of
Helsinki, P.O. Box 64, Helsinki, 00014 University of Helsinki
[2]Climate & Atmosphere Research Centre (CARE-C), The Cyprus Institute, P.O. Box 27456, Nicosia, CY-
1645, Cyprus
[3]Finnish Meteorological Institute, Helsinki, Finland
*correspondence to t.jokinen@cyi.ac.cy
**Abstract:**
Aerosol particles form in the atmosphere by clustering of certain atmospheric vapors. After growing to larger
particles by condensation of low volatile gases, they can affect the Earth's climate directly by scattering light
and indirectly by acting as cloud condensation nuclei. Observations of low-volatility aerosol precursor gases
have been reported around the world but longer-term measurement series and any Arctic data sets showing
seasonal variation are close to non-existent. In here, we present ~7 months of aerosol precursor gas
measurements performed with the nitrate based chemical ionization mass spectrometer (CI-APi-TOF). We
deployed our measurements ~150 km North of the Arctic Circle at the continental Finnish sub-Arctic field
station, SMEAR I, located in Värriö strict nature reserve. We report concentration measurements of the most
common new particle formation related compounds; sulfuric acid (SA), methane sulfonic acid (MSA), iodic
acid (IA) and a total concentration of highly oxygenated organic compounds (HOMs). At this remote
measurement site, SA is originated both from anthropogenic and biological sources and has a clear diurnal
cycle but no significant seasonal variation. MSA shows a more distinct seasonal cycle with concentrations
peaking in the summer. Of the measured compounds, iodic acid concentrations are the most stable throughout
the measurement period, except in April, when the concentration of IA is significantly higher than during the
rest of the year. Otherwise, IA has almost identical daily maximum concentrations in spring, summer and
autumn, and on new particle formation event or non-event days. HOMs are abundant during the summer
months and low in winter months. Due to the low winter concentrations and their high correlation with ambient
air temperature, we suggest that most of HOMs are products of biogenic emissions, most probably
monoterpene oxidation products. New particle formation events at SMEAR I happen under relatively low
temperatures with a fast temperature rise in the morning followed by decreasing relative humidity during the
day. The ozone concentrations are on average ~10 ppbv higher on NPF days than non-event days. During NPF
days, we have on average higher SA concentration peaking at noon, higher MSA concentrations in the
afternoon and slightly higher IA concentration than during non-event days. All together, these are the first long
term measurements of aerosol forming vapors from the SMEAR I in the sub-arctic region, and the results help
us to understand atmospheric chemical processes and aerosol formation in the rapidly changing Arctic.
**Introduction:**
The climate of sub-Arctic region is characterized with some of the most extreme temperature variations on
Earth. We expect that during the course of the 21[st] century, the boreal forest is to experience the largest increase
in temperatures of all forest biomes (IPCC, 2013), making it the most vulnerable to climate change. The boreal
forest (taiga) covers most of the sub-Arctic and encompasses more than 30% of all forests on Earth, being one
of the largest biome in the world (Brandt et al., 2013). The expected rate of changes, may overwhelm the
resilience of forest ecosystems and possibly lead to significant biome-level changes (Reyer et al., 2015). The
forest-atmosphere systems are closely interlinked to one another. The forest stores carbon and water in the



peat, soil and as biomass while at the same time vegetation emits volatile organic compounds (VOC) into the
atmosphere (Bradshaw and Warkentin, 2015). In the Arctic, summer is short, but solar radiation is abundant
and extends the daylight hours all the way to midnight and beyond. On the other hand, during the polar night
air pollutants accumulate in the atmosphere due to cold and stable atmosphere, while turbulent mixing is
inhibited, and the lack of removal processes lead to the formation of Arctic haze (Stohl, 2006).  These features
make the Arctic an interesting study region for photochemistry of reduced atmospheric compounds. Oxidation
processes that dominantly occur in the summer time control the processes removing VOCs and other traces
gases, such as $SO_2$ and $NO_x$, from the atmosphere in the Arctic. Detailed understanding of atmospheric
processes leading to aerosol precursor formation and gas-to-particle conversion and their role in feedback
mechanisms help in assessing the future climate.
Aerosol and trace gas measurements in the sub-Arctic field station SMEAR I, go back to the 90s (Ahonen et
al., 1997; Kulmala et al., 1998; Mäkelä et al., 1997). Trace gas and aerosol measurements at SMEAR I started
in 1992 making them one of the longest continuous measurements of aerosol particle number and size
distributions in the Arctic (Ruuskanen et al., 2003). These long-term measurements show that aerosol particles
regularly form and grow from very small sizes (< 8 nm diameter) with the highest frequency in the spring,
between March and May (Dal Maso et al., 2007; Vehkamäki et al., 2004). It is suggested, that spring promotes
new particle formation (NPF) because of the awakening of biological processes after the winter. At SMEAR I
the snow only melts away in May-June and thus, many biological processes (photosynthesis) activate while
the snow is still deep. This makes the Arctic spring a very complex environment for atmospheric chemistry
with possible emission sources from melting snow, ice, melt water, vegetation and transport from other areas.
At SMEAR I, most of the observed NPF events are either connected to clean air arriving from the Northern
sector (originating from The Arctic Ocean and transported over boreal forest, Dal Maso et al., 2007) or the
polluted air masses from the Eastern sector  (Kyrö et al., 2014; Sipilä et al., 2021). Annually, around 30-60
NPF events are recorded at SMEAR I, of which around half could be initiated by anthropogenic air pollutants
from the Kola Peninsula (Kyrö et al., 2014; Pirjola et al., 1998; Sipilä et al., 2021) leaving half of the events
occurring from natural sources. The trend of NPF occurrence in Värriö is decreasing, as the anthropogenic
sulfur dioxide emissions are decreasing in Russia (Kyrö et al., 2014).
Formation and growth of new particles at SMEAR I usually happen during daylight, highlighting the
importance of photochemical activities. However, unlike most other locations, NPF is also observed during
nighttime or polar night (Kyrö et al., 2014; Vehkamäki et al., 2004). Formation and growth processes of
aerosols seem not to be correlated with each other at SMEAR I (Vehkamäki et al., 2004). Earlier literature
reports, that the formation rate (J) has no clear seasonal trend, while the growth rates (GR) of small particles
clearly peak during summer (Ruuskanen et al., 2007). This indicates that different chemistry drives the initial
cluster formation and the subsequent growth processes. From the observed nucleation rates it has been
proposed that NPF at SMEAR I could be due to sulfuric acid –ammonia (-water) nucleation (Napari et al.,
2002) likely dominated by ion-induced channel at least during winter months (Sipilä et al., 2021). Kyrö et al.,
2014 concludes that 20-50% of the condensational growth can also be explained by sulfuric acid in Värriö.
Other studies speculate about the possibility of different organic compounds participating in NPF in the sub-
Arctic. Tunved et al., 2006 studied the air masses arriving to SMEAR I station and concluded that the aerosol
mass increased linearly with time that the air masses travelled over land. The concentration of condensing
gases over the boreal forest was concluded to be high and most likely consisting mainly of oxidation products
of terpenes (VOCs) that are emitted by the forest. At SMEAR II station in Hyytiälä, approximately 700 km
South-West of Värriö, oxidized organics mostly explain the growth of newly formed particles (Bianchi et al.,
2017; Ehn et al., 2014). However, direct measurements of the aerosol forming and growing vapor species are
still lacking from SMEAR I except during wintertime without biogenic activity when sulfuric acid has been
shown to be primarily responsible on formation and growth (Sipilä et al., 2021). In Värriö, the role of NPF is
critical in forming of cloud condensation nuclei (CCN), since measurements show that the number of CCN
can increase up to 800 % as a result of NPF (Kerminen et al., 2012). In other locations in the boreal forest and
Arctic, some measurements shed light into the possible chemical components that could be forming particles
in Värriö. Currently, the closest continuous measurements with the nitrate based CI-APi-TOF are conducted



in Hyytiälä at the SMEAR II-station (Jokinen et al., 2012, 2017; Kulmala et al., 2013). In Hyytiälä there is
direct evidence on the key role of the photochemical production of sulfuric acid and HOMs maintaining
atmospheric NPF (Bianchi et al., 2017; Ehn et al., 2014; Jokinen et al., 2017; Kulmala et al., 2013).
Other chemical composition measurements of aerosol precursors have been conducted only in a few locations
in the High-Arctic and over the Arctic Ocean (Baccarini et al., 2020; Beck et al., 2021; He et al., 2021; Sipilä
et al., 2016). These studies show that in the Arctic, the marginal ice zone and the coast of the Arctic Ocean is
a source of atmospheric iodic acid that is efficiently forming new particles. Sulfuric acid and MSA
concentrations were also reported (Beck et al., 2021), but they were much lower in concentration than iodic
acid (Baccarini et al., 2020). However, the chemistry behind NPF is not that simple, even the pristine Arctic
air. The clean air above the Arctic Ocean is abundant in dimethyl sulfide (DMS) emitted by phytoplankton,
that rapidly oxidizes into sulfuric acid and MSA on sunny days and consequently forms cloud condensation
nuclei (Charlson et al., 1987; Park et al., 2018). Beck et al., (2021) report, that in Svalbard in the Arctic Ocean,
sulfuric acid and methane sulfonic acid contribute to the formation of secondary aerosol. They also observed
that these compounds formed particles large enough to contribute to some extent to cloud condensation nuclei
(CCN). This is supported by  measurements of aerosol chemical composition from the Arctic that commonly
report MSA in particulate matter (Dall Osto et al., 2018; Kerminen et al., 1997). According to Beck et al.
(2021) the initial aerosol formation in the high Arctic occurs via ion-induced nucleation of sulfuric acid and
ammonia and subsequent growth by mainly sulfuric acid and MSA condensation during springtime and highly
oxygenated organic molecules (HOM) during summertime. By contrast, in an ice-covered region around
Villum, Greenland, Beck et al. (2021) observed new particle formation driven by iodic acid, but the particles
remained small and did not grow to CCN sizes due to insufficient concentration of condensing vapors. Since
the Arctic CCN number concentrations are low in general, formation of new particles is a very sensitive process
affecting the composition of the aerosol population and CCN numbers in the area. Also in Värriö, the role of
NPF is critical in forming of cloud condensation nuclei (CCN), since measurements show that the number of
CCN can increase up to 800 % as a result of NPF (Kerminen et al., 2012).
In this article, we present the measurements of aerosol precursor molecules from the continental SMEAR I
station, ~150 km North of the Arctic Circle and ~150 km from the Arctic Ocean. We measured sulfuric acid,
methane sulfonic acid, iodic acid and highly oxygenated organic compound concentrations with a sulfuric acid
calibrated CI-APi-TOF (Jokinen et al., 2012; Kürten et al., 2012) to determine their levels in the sub-Arctic
boreal forest and to understand whether these species are connected with the aerosol formation process in the
area.
**Methods, measurement site and instrumentation:**
The core of this work is measurements of gas phase aerosol precursors. We use the nitrate chemical ionization
atmospheric pressure interface time-of-flight mass spectrometer (CI-APi-ToF) that has been operational at the
SMEAR I-station (N67°46, E29°36) in Eastern-Lapland since the early spring of 2019. SMEAR stands for
Station for Measuring Ecosystem – Atmosphere Relations. Measurements were done on top of Kotovaara hill
(390 m a.s.l.), close to ground level in an air-conditioned small log wood cottage. The cottage is surrounded
by ~65-year-old Scotts pine forest. More details about the station can be found in earlier publications (Hari et
al., 1994; Kyrö et al., 2014). The mass spectrometric measurements are designed to start a long-term
measurement series of atmospheric aerosol forming trace gases in the Finnish Lapland and the measurements
are ongoing to this day. We measure e.g. sulfuric acid, iodic acid, highly oxygenated organic molecules and
methane sulfonic acid with high time resolution and precision. The measurements are running in Finnish winter
time (UTC+2) throughout the year.
We calibrated the CI-APi-TOF twice during the measurement period and run the instrument with the same
settings for the whole measurement period reported in this paper. We calibrated the instrument using a sulfuric
acid calibrator described in Kürten et al., 2012. The calibration factor from the two separate calibrations were
1) $7 \cdot 10^9$ and 2) $8 \cdot 10^9$ and we use the average $7.5 \cdot 10^9$ in our study calculate the concentrations of all reported
compounds. This factor includes the loss parameter due to the ~1 m long unheated inlet tube (3/4" stainless



steel). HOMs and iodic acid have been estimated to be charged similarly at the kinetic limit as sulfuric acid
(Ehn et al., 2014; Sipilä et al., 2016), so the calibration factor for them should be similar, but please note, that
the concentration of other compounds than SA can be highly uncertain due to different ionizing efficiencies,
sensitivities and other unknown uncertainties. The sulfuric acid, iodic acid and MSA data presented in this
study are all results of high-resolution peak fitting of the CI-APi-TOF, in order to avoid inaccurate
identification of compounds and to separate overlapping peaks. The HOM data is a sum of mass-to-charge
ratios from 300 to 400 Th, representing the monomer HOM range ($C_{10}$ compound range), 401 to 500 Th for
the slightly larger HOMs ($C_{15}$ compound range) and 501 to 600 Th for the dimer species ($C_{20}$ compound range).
We also give the sum of these all (from 300 to 600 Th). The goal of this article is not to specify different HOM
compounds or to study NPF in mechanistic details but to give an overview of general seasonal trends and
variations of these selected species. Note that since this is a sum of all peaks in the selected mass range, we
cannot assure that all the compounds included are HOMs. However, the investigation in laboratory conditions
show that the nitrate-CI-APi-TOF is highly selective and sensitive towards HOMs with $O > 5$ (Riva et al.,
2019) and with hydroperoxide (-OOH) functionalities (Hyttinen et al., 2015). All data obtained from the CI-
APi-TOF we analyzed using tofTools program described in (Junninen et al., 2010) and averaged over an hour.
The original data time resolution is 5 sec. The uncertainty range of the measured concentrations reported in
this study is estimated to be $-50\%/+100\%$ and the limit of detection, LOD $4 \cdot 10^4$ molecules cm$^{-3}$ (Jokinen et
al., 2012).
To classify NPF events recorded during the measurement period, we used the data measured by a Differential
Mobility Particle Sizer (DMPS). The DMPS instrument and earlier statistics of NPF events in Värriö has been
documented by (Dal Maso et al., 2007; Vana et al., 2016; Vehkamäki et al., 2004). The NPF events were
classified according to (Dal Maso et al., 2005). Total aerosol particle number concentration was measured with
a Condensation Particle Counter (CPC, TSI 3776) in the size range of 3 – 800 nm. Air ion size distributions
were measured with the Neutral cluster and Air Ion Spectrometer, NAIS (Kulmala et al., 2007; Manninen et
al., 2016; Mirme and Mirme, 2013) that measures negative and positive ions in the size range of 0.8 – 42 nm
in mobility diameter and total particle size distribution between ~2 and 42 nm. All meteorological parameters,
trace gas concentrations and aerosol data we downloaded directly from smartSMEAR open access database
(https://smear.avaa.csc.fi/) and all mass spectrometric data are available on request.
**Results and discussion:**
**a. Overview of the whole measurement period:**
You can see a time series of the most common aerosol precursor compounds; sulfuric acid, methane sulfonic
acid, iodic acid and sums of different HOM groups in Figure 1. This figure depicts the whole measurement
period from April 4 to October 27 in 2019. Overall, we succeeded to measure the whole 7 month period almost
uninterruptedly. Only a few short power cuts stopped our measurements during this time. Iodic acid data is
missing from late July since its peak could not be separated well enough from overlapping peaks in the spectra
during this time. This was due to poor resolution (low signal of IO$_3^-$ close to another peak) that makes peak
integration to give negative, unreal values and we thus decided to flag them out. After late October, the
instrument malfunctioned and stopped our measurements. In this particular article, we present data from spring
(Apr-May), summer (Jun-Jul-Aug) and autumn (Sep-Oct) 2019. More about the SMEAR I winter observations
can be read in Sipilä et al., 2021 were they report observations of polar night pollution events from Värriö after
the CI-APi-TOF was fixed.

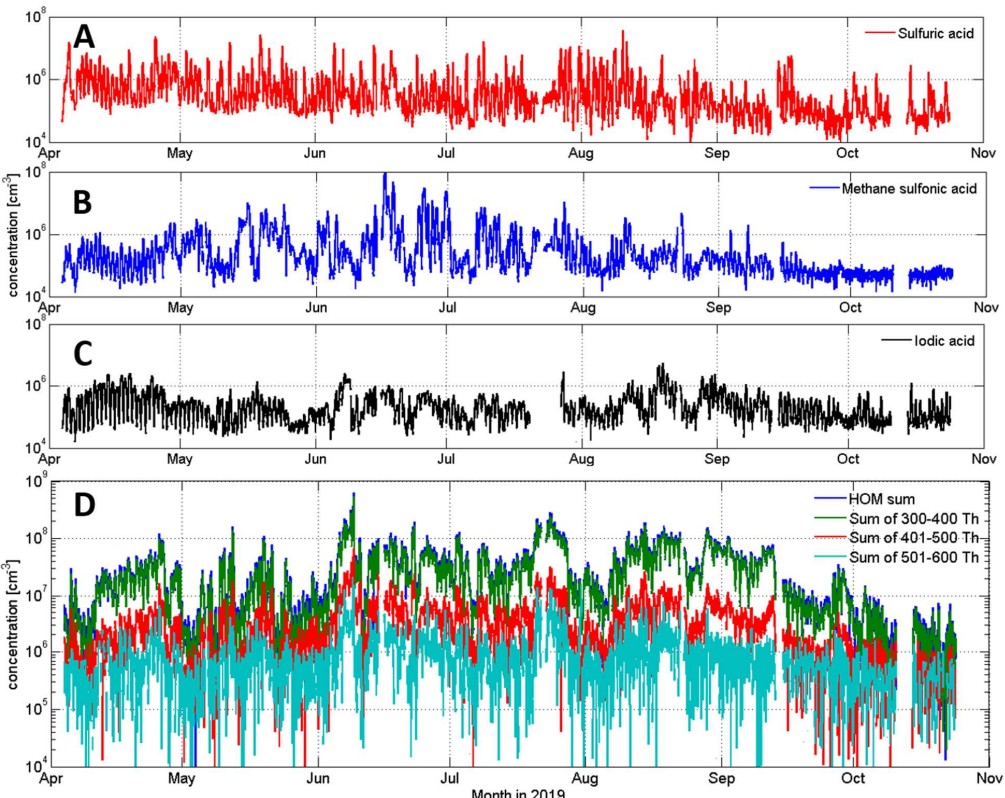


**Figure 1.** Overview of sulfuric (A), methane sulfonic (B) and iodic acid (C), as well as HOM (D) concentrations at SMEAR I in April to October 2019. All data in panels A-C are resulting from high-resolution peak fitting. HOM data are sums of certain mass ranges; from 300 to 400 Th in green, representing C10 or HOM monomer compounds, from 401 to 500 Th in red, representing C15 compound and from 501 to 600 Th on light blue, representing C20 or HOM dimer compounds. The sum of HOM (blue) is a sum of the aforementioned mass ranges. The sum of HOMs is approximately one order of magnitude higher than SA, MSA or IA concentrations during this measurement period.

In Figure 2 we show some of the most interesting environmental and meteorological parameters that influence the atmospheric gas composition during the measurements period; temperature, global radiation and snow depth (Figure 2A). There are some special features in year 2019; the summer had two heat waves, when the air temperature rose up to 29.2 °C in early June and to almost the same values in late July. These episodes are getting more common in Lapland due to climate change. These warmer conditions will probably change the emissions of trace gases including the composition and abundance of aerosol precursors in the future Arctic environment (Schmale et al., 2021).

From Figure 2, we see that the snow covered period ended in 2019 in late May and snow started to accumulate again in mid-October. Solar radiation (Figure 2A) is intense in Värriö during springtime and gives Värriö favorable photo-oxidizing conditions, effectively removing air pollutants and trace gases from the atmosphere. Photochemical activity produces ozone in springtime and this is visible in very high ozone concentrations at the site (Figure 2B). Ozone concentrations were around 55 ppbv in April and decreased to ~30 ppbv in the late summer and autumn (Figure 2B). The spring ozone concentration in 2019 was significantly higher than the





previous reports from the years 1992 to 2001, when monthly mean concentrations of ozone varied between
25-40 ppbv (Ruuskanen et al., 2003).

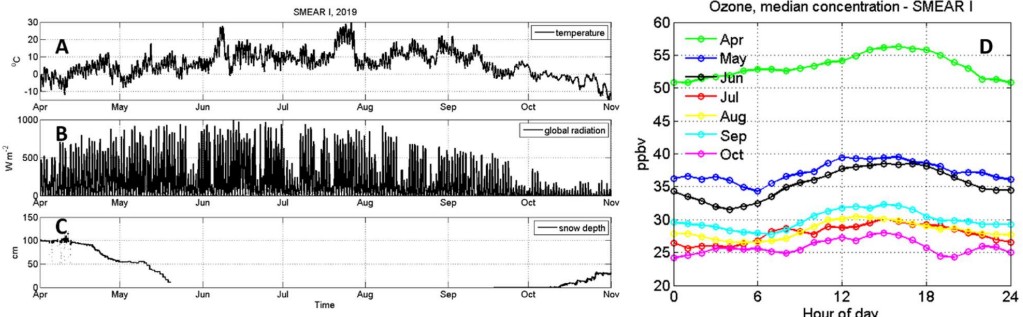


**Figure 2:** Observations of temperature (A), global radiation (B) and snow depth (C) at SMEAR I during the
measurement period. Monthly median ozone concentration in ppbv (D), is showing the relatively high level of
ground level ozone during springtime (Apr).
The springtime diurnal solar cycle is clearly visible with all studied compounds. All measured aerosol
precursor compounds are abundant even during the period when snow covers the ground in the spring. The
HOM concentrations follow the increasing solar radiation and rising temperature. MSA has a stronger diurnal
cycle before the snow melt than after it. This may be due to rain and cloudy conditions that are more common
in the summer. Sulfuric acid and iodic acid do not have such strong seasonal variation than HOMs and MSA.
The aerosol precursor concentrations are discussed in more detail in the following sections.
**Seasonal and monthly variation of SA, MSA, iodic acid and HOM concentrations**
We present the diurnal variation of aerosol precursors; sulfuric acid, methane sulfonic acid, iodic acid and
highly oxygenated molecule, concentrations separately for different seasons in Figure 3. Strong seasonality is
most evident in sulfuric acid and HOM concentrations. SA is at its highest in the spring, decreasing toward
summer and autumn while HOMs reach their maximum in the summer. The increase in HOMs in the summer
at SMEAR I is linked to the increased emissions of VOCs from vegetation that oxidize into HOMs via
ozonolysis (Ehn et al., 2014) and OH-radical reactions (Berndt et al., 2016; Jokinen et al., 2014, 2017; Wang
et al., 2018). The overall lowest aerosol precursor concentrations we detect during autumn (winter data was
missing from this study, see Sipilä et al. 2021, for winter time observations made promptly after the period
reported here). MSA shows very similar concentrations during spring and summer, and drops down to the limit
of detection level for autumn. Iodic acid acts very differently than the other compounds. We observe iodic acid
to have a similar level of concentration throughout the measurement period and the concentration almost seem
to "saturate" during daylight hours. This daytime maximum stays at the same level about 5 hours longer during
spring than in the autumn. The day length getting shorter towards the autumn explains this behavior. The
maximum hourly median concentrations for the measured compounds are ~2 · $10^6$ cm$^{-3}$ for SA (spring), ~5 ·
$10^5$ cm$^{-3}$ for MSA (summer), ~3· $10^5$ cm$^{-3}$ for iodic acid (all seasons) and ~5 · $10^7$ cm$^{-3}$ for the sum of HOMs
(summer, mass range from 300 to 600 Th).
We can compare these numbers to SMEAR II long-term (5-year median concentration) observations, were the
median peak SA concentrations are ~1.5 · $10^6$ cm$^{-3}$, ~1 · $10^6$ cm$^{-3}$ and ~3· $10^5$ cm$^{-3}$ for spring, summer and
autumn, respectively (Sulo et al., 2021). These measured concentrations are very similar to SMEAR I
observations except a slightly higher summer and autumn SA concentration at SMEAR II. However, it should
be noted that the springtime measurements from SMEAR I do not include March data, which makes the
springtime comparison uncertain. There is also a difference in the timing of the peak SA concentration in the
summer. At SMEAR I the peak concentration is reached at noon and at SMEAR II it can be found some hours





earlier, already around eight o'clock in the morning (Sulo et al., 2021). In the case of HOMs, we cannot
compare the concentrations directly to Sulo et al. (2021) as they calculated the sum of HOMs differently, only
taking into account the most abundant signals and separating nitrate and non-nitrate HOMs. However, we take
the liberty to compare diurnal and seasonal variations. Both at SMEAR I and II, observations show the highest
HOM concentrations during summer, while the autumn concentrations areone order of magnitude lower. The
comparison between these sites reveals a different diurnal variation of HOMs. At SMEAR I, the HOMs have
a maximum around noon, spanning to the afternoon (Figure 3). At SMEAR II, HOMs have two maxima, one
at noon and another one in the early evening. From these, the latter is connected to non-nitrate monomer and
dimer HOMs and nitrate dimer HOMs. At SMEAR I the lack of an evening maximum could indicate that
HOM dimer formation is less dominant at SMEAR I compared to SMEAR II due to lower air temperatures,
or due to the different diurnal cycle of oxidants due to longer hours of solar radiation North of the Arctic Circle.

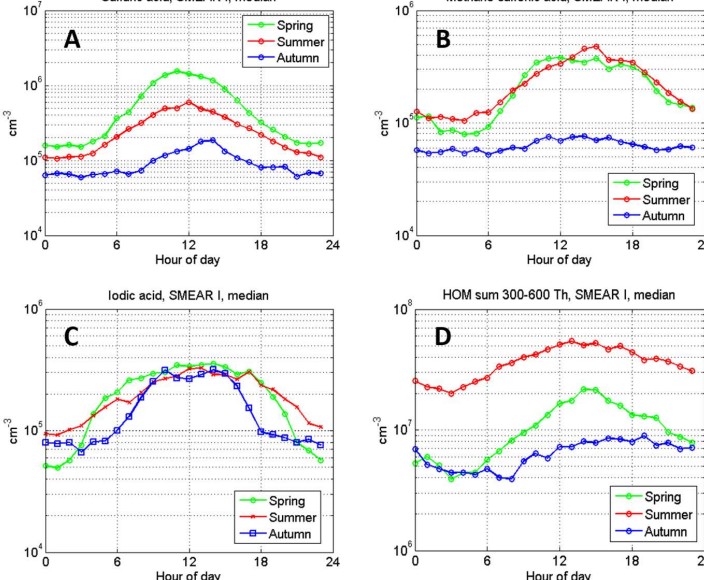

**Figure 3.** Diurnal variation of aerosol precursor gas median concentrations in different seasons: A) sulfuric
acid, B) methane sulfonic acid, C) iodic acid and d) the sum of HOMs in the 300 to 600 Th mass range.
When analyzing the monthly aerosol precursor profiles in Figure 4, we observe that the springtime atmosphere
is abundant in SA and iodic acid that have the highest median concentrations in April. MSA and HOMs
concentrations peak in June. The MSA behavior is likely connected to the algae blooms in the Arctic Ocean
that peak around midsummer. The marine emissions of DMS oxidize in the atmosphere to sulfur dioxide,
sulfuric acid and to MSA (e.g. Park et al., 2018). However, sulfuric acid has more sources, since $SO_2$, has also
anthropogenic sources. At SMEAR I we cannot distinguish these sources precisely (more discussion about this
in section 3.3.). It is notable that the peak concentration of MSA is earlier in the day in April, around 12 o'clock
noon, than it is later in the year when the peak concentration is reached in the late afternoon (from 13:00 to
18:00 o'clock). There are no previous MSA concentration reports from the SMEAR stations but some gas
phase MSA results from Antarctica show maximum of $1 \cdot 10^5$ cm$^{-3}$ to $1 \cdot 10^7$ cm$^{-3}$ concentrations (Mauldin et
al., 2004, Mauldin et al., 2010, Jokinen et al., 2018). In the Arctic, around half a year measurement series from
Villum in Greenland show MSA concentrations <$10^6$ cm$^{-3}$ (Mar – Sep) and from$10^5$ cm$^{-3}$ to $10^7$ cm$^{-3}$ with the
highest concentrations in June in Ny-Ålesund (Beck et al., 2021). Our measurements from the SMEAR I fall
in between these extremes.



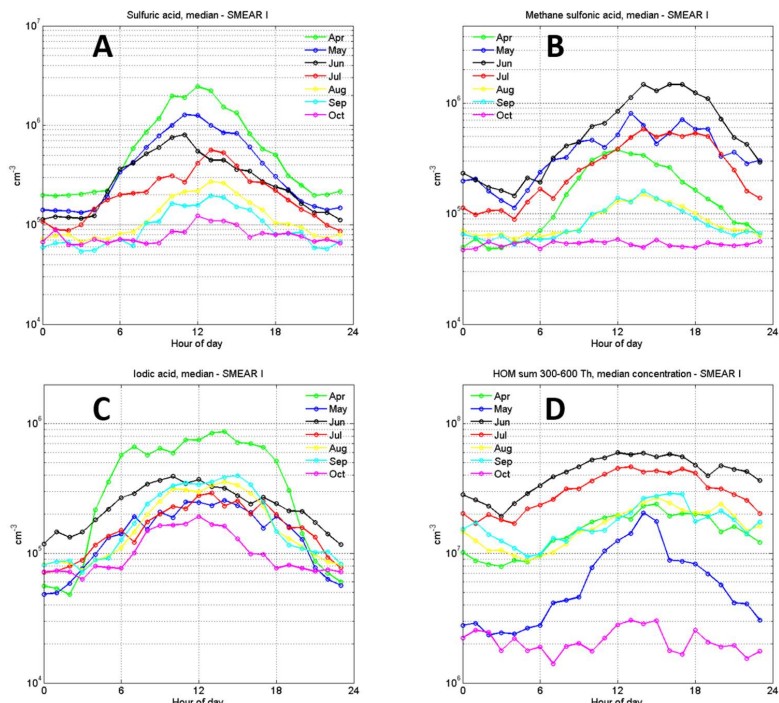

**Figure 4.** Monthly median concentrations of A) sulfuric acid, B) methane sulfonic acid, C) iodic acid and d) the sum of HOMs in the 300 to 600 Th mass range.

These are also the first reported results of iodic acid measurements from SMEAR I and they represent a continental location, the White Sea coast being ~130 km South East and the Barents sea ~230 km to the North East. Iodic acid, iodine and iodic oxoacid emissions are commonly connected to coastal or marine environments (Baccarini et al., 2020; McFiggans et al., 2010; O'dowd et al., 2002; Sipilä et al., 2016; Yu et al., 2019) due to the fact that the ocean surface is a major source of iodine (Carpenter et al., 2013). While it is not precisely known how iodic acid forms in the gas phase, its formation requires oxidation of the initial precursors ($IO_x$ species) by ozone and the last steps of its formation is potentially driven by reaction with OH (Chameides and Davis, 1980).

Compared to the other precursor compounds, iodic acid has the most stable concentration between seasons, with a long increasing period in April during the snow-melting season. This is likely due to the simultaneously increasing ozone concentrations (Fig 2B) and solar radiation. In contrast to measurements from the Arctic Ocean (Baccarini et al., 2020), we did not observe a clear increase in iodic acid concentration in the autumn due to freezing. We find that September had only marginally higher concentrations compared to August or July (Fig 4). Winter measurements would be necessary to estimate the effect of freezing in the concentration of IA.

The source of iodic acid on a continental site like the SMEAR I is an interesting subject to speculate. The observed $HIO_3$ peak in April could indicate that there could be an influence from air masses exposed to Arctic marine environment, due to ocean surface acting as a major source of atmospheric iodine (Carpenter et al., 2013). The increasing temperature in the spring induce a higher activity of phytoplankton in the nearby Barents Sea and Norwegian Sea that remains ice free, even during the winter, and could result in the higher emission of precursors for iodic acid (Lai et al., 2011). Higher temperature would also result in more efficient advection, which would transport species faster from emission points to SMEAR I. The calculated back trajectories



support the idea that iodine-rich air masses arrive from the West or northwest to SMEAR I (discussed in details
in section 3.3. New Particle Formation evens and Figure 10). This would be the hypothesis of the long-range
transport for source of iodic acid in SMEAR I. On the contrary, the strong diurnal variation on iodic acid
concentration seen as one order of magnitude difference between noon and midnight, suggests fast on site
chemistry, which is not consistent with long-range transport. Also iodic acid life time against condensational
loss is expected to be short, in the range of an ~hour, this suggests that the source of $HIO_3$ is close to or at the
site of measurements. Land vegetation is a source of methyl iodide ($CH_3I$) that could be the source of iodic
acid at SMEAR I, at least during summer (Sive et al., 2007).
Most interestingly, we seem to have an emission source of iodine during all seasons. There are no reports on
iodine emissions from continental snow, but we hypothesize that one possible source of iodine in SMEAR I
during spring is the melting snowpack. This is possible due to the deposition of sea salts on snow particularly
during dark periods that activate during the spring and are re-emitted to the atmosphere through heterogeneous
photochemistry of iodide, and iodate ions (Raso et al., 2017; Spolaor et al., 2019). There are also possible
forest emissions of iodinated organics, similar to New England growing season (Raso et al., 2017), that might
be enhanced by higher temperature or high ozone concentrations. This type of emissions of iodinated gases,
or their implications, have not been studied before but these observations might direct research into emission
studies at SMEAR I, since our findings indicate that vegetation could be an emission source of iodine.
The sum of HOMs in SMEAR I reaches up to a median ~5 · $10^7$ cm$^{-3}$ concentration in the summer. This is
about one order of magnitude lower than the concentrations reported from the SMEAR II station in Hyytiälä
(Yan et al., 2016), about 700 km south, where HOMs are at a maximum of ~6 · $10^8$ cm$^{-3}$ during spring daytime.
It is striking how well the concentration of HOMs follow the air temperature (Figure 5). From the temperature
dependency, we can speculate that most VOCs emitted by vegetation close to Värriö could be monoterpenes
due to their strong temperature dependency. This is supported by emission rate measurements of VOCs
showing that in northern Finland 60 to 85 % are accounted by α- and β-pinene emissions (Tarvainen et al.,
2004). However, sesquiterpene emissions from nearby wetlands could contribute to HOMs since their
emissions are also temperature dependent and they are emitted by the boreal wetlands (Hellén et al., 2020;
Seco et al., 2020). As HOMs are oxidation products of VOCs, it is evident that the HOM concentration will
increase in SMEAR I in the future with the increasing VOC emissions, including isoprene, monoterpenes and
sesquiterpenes, due to temperature rise (Ghirardo et al., 2020; Tiiva et al., 2008; Valolahti et al., 2015).

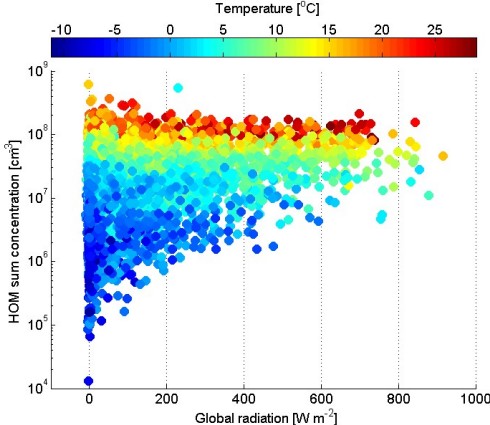


**Figure 5.** HOM concentration (cm$^{-3}$) measured at SMEAR I (sum of mass range from 300 to 600 Th) as a
function of global radiation (W m$^{-2}$). The color in the plot represents air temperature in °C. The plot includes
all data measured from April to October 2019.

### New particle formation events;

During the measurement period from 4 April 2019 to 27 October 2019, we observed 36 regional NPF events
in total and our CI-APi-TOF data covers 33 of these NPF days. During the same period, we observed 75 non-
event days without clear signs of particle formation (Dal Maso et al., 2005). Rest of the days during our
measurement period were defined as undefined, bad data or partly bad data days and these were excluded from
our analysis. In this chapter, we focus on trace gases, meteorological parameters and aerosol precursor detected
gases during NPF days and compare them to non-event days.
We plot NPF and non-event days median average number size distribution of aerosol particles (from 3 to 800
nm) in Figure 6, and the total number concentration and the 2-7 nm air ion concentrations in Figure 7. In figure
6, in the case of NPF event days we see a distinct "banana" plot, where small < 10 nm, particles are forming
and growing with time. The DMPS data is plotted from 2.82 nm to 708 nm but note that the channels below
~5 nm have much larger uncertainties than those above. The median event start time is located around noon
and the growth of particles continues steadily until midnight. However, when looking at individual days, there
is a large variation in the start-times of the particle formation, some events start early in the morning or even
in the night, while some start in late afternoon. Non-event days show very few particles in the < 10 nm size
bins.

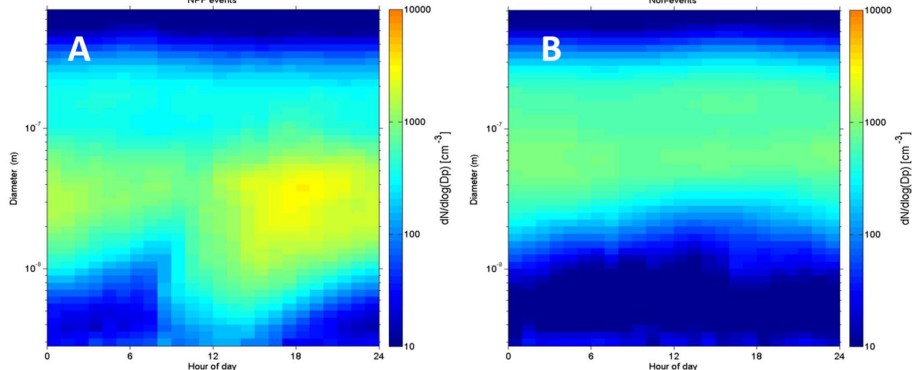


**Figure 6.** This figure depicts the median number size distribution during all observed during NPF events (n =
33) and non-events (n = 75) during our measurement period. The data is collected with a DMPS and size bins
from 2.82 to 708 nm are plotted.

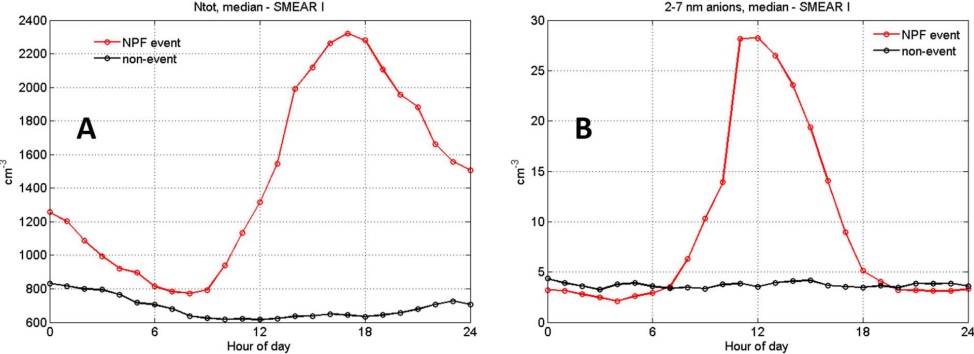





**Figure 7.** Median total particle concentration (Ntot) in A) and 2-7 nm negative ion concentrations in B) at
SMEAR I during NPF event (red, n = 33) and non-event days (black, n = 75). The total particle number
concentration is recorded with a CPC and air ion concentrations with a NAIS.
The total number of particles measured at the site during NPF events rises up to ~2400 cm$^{-3}$ reaching the
maximum concentration at ~17 o'clock in the evening. This shows that NPF is an important source of aerosol
particles in Värriö as previously reported (Vehkamäki et al., 2004). Non-event days have clearly lower particle
concentrations throughout the day, staying lower than 1000 cm$^{-3}$ on average. The measured 2-7 nm anion
concentrations stay very low during non-event days. As intermediate ions form mainly during NPF, their
concentrations are used as indicator of NPF events in boreal environments (Leino et al., 2016). On NPF days,
we see a peak in the anion concentration at noon, the concentration being about six times higher than during
non-event days. This indicates that negative ions may play a role in SMEAR I particle formation events.
Figure 8 shows the differences in temperature, relative humidity, global radiation and ozone concentration
between NPF event days (in red) and non-event days (black). In Värriö, NPF events preferably happen in
relatively low temperatures with a fast temperature rise in the early morning hours, lower and decreasing RH
during the NPF days compared to non-event days. NPF days have clearly higher global irradiance values and
about 10 ppbv higher ozone concentrations than non-event days. The meteorological conditions favoring NPF
are thus similar than at the SMEAR II station in Hyytiälä, where sunny clear sky days with low RH and
condensation sink along with wind directions from the cleaner northerly sector are forecasting NPF events
(Nieminen et al., 2014).

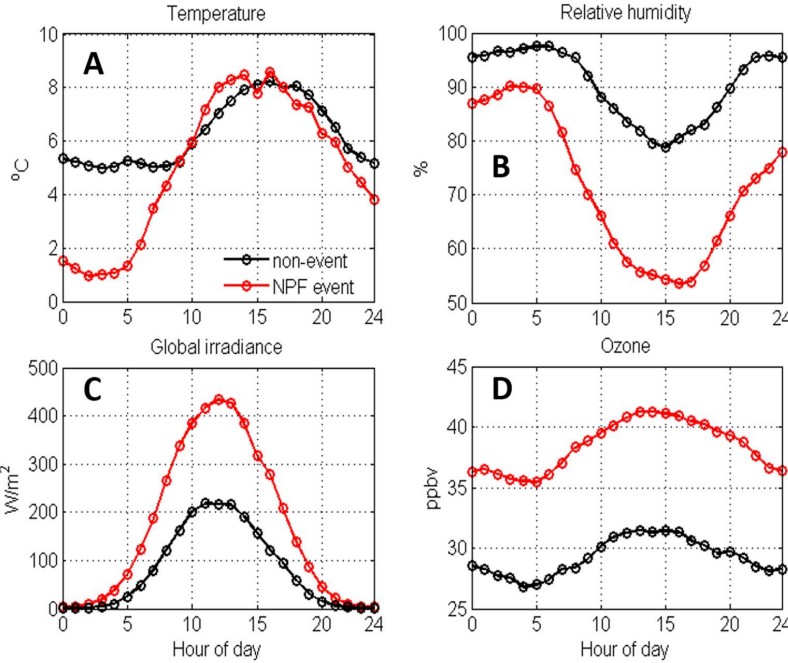


**Figure 8.** Average temperature (°C) in panel A), relative humidity (%) in B), global radiation (W m$^{-2}$) in C)
and ozone concentration (ppbv) in D), all measured at SMEAR I during NPF event (red, n = 33) and non-event
days (black, n = 75).
Next, we show the concentrations of aerosol precursor compounds during NPF and non-event days in figure
9. (Kulmala et al., 2013)The sulfuric acid concentrations closely follow the solar irradiation profile (Figure 8).



Similarly to the results obtained from the high Arctic, Svalbard, also MSA is elevated during NPF events,
especially during summer, and could possibly contribute aerosol growth (Beck et al., 2021). We observe close
to an order of magnitude higher MSA concentration between the events and non-events days, highlighting the
dominant role of sulfur species to nucleation and growth in general at this site. In order to attribute the source
of sulfur species and IA during the event and non-event days we performed a cluster analysis using a
geographical information system (GIS) based software, Trajstat (Wang et al., 2009). The NCEP/NCAR
reanalysis data was used as meteorological input for the model (Kalnay et al., 1996). The simulations were
performed at an arrival height of 250 m. a.g.l. SMEAR I station is located approximately at similar height (390
m a.s.l), thus representing the air masses arriving at the station even during strong temperature inversions
(Sipilä et al., 2021).

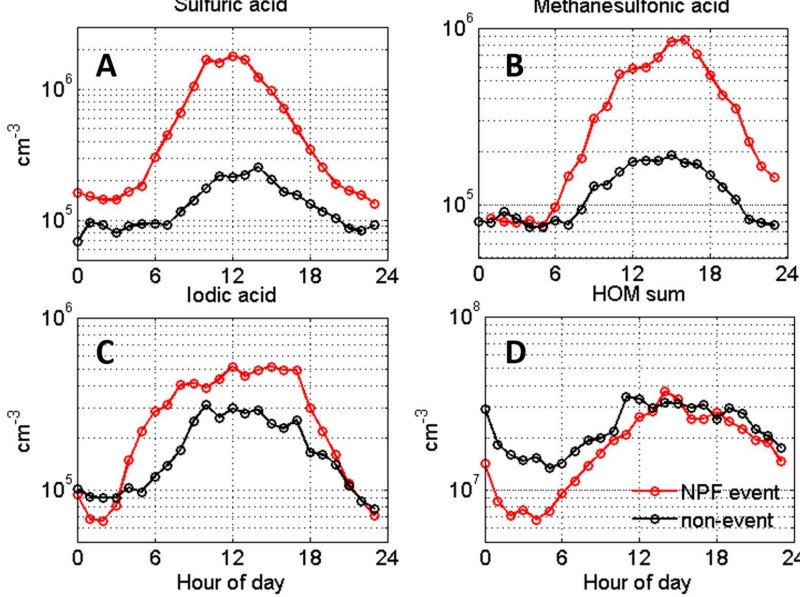


**Figure 9.** Aerosol precursor gases in SMEAR I during NPF (red, n = 33) and non-event days (black, n = 75).
The data is hourly median average.
Higher concentrations of aerosol precursors SA, MSA and IA are connected to the air masses that arrive to
SMEAR I from the Arctic Ocean (Figure 10). Cluster analysis of air mass back trajectories arriving to Värriö
during NPF days clearly shows, that most NPF events occur when the air mass was exposed to marine
environments within the last 72 hours. In our case, mainly the Norwegian Sea in the West (58 %) or the Barents
(21 %) and Kara Seas (21 %) in the Arctic Ocean. This seems relevant to our results since the marine
environment in the North is emitting large amounts of dimethyl sulfide (DMS), a precursor for SA and MSA
(Levasseur, 2013) and iodine species that further oxidize to IA (Baccarini et al., 2020; Sherwen et al., 2016).
A fraction of air masses that are connected to both NPF (21 %) and non-event days (37 %) are coming to
SMEAR I from the Kola peninsula that is connected to high SO$_2$ emissions, higher particle number
concentrations and winter time NPF events (Sipilä et al., 2021). Most non-event air masses arrive to Värriö
from South-West (49 %) crossing northern Finland and Sweden.
In addition from Figure 9 we observe that we cannot rule out the contribution of iodic acid in NPF in SMEAR
I, but with the recorded concentration, it usually is not enough to initiate NPF (He et al., 2021). Although iodic
acid concentrations are slightly larger on NPF days than non-event days, the rise in concentration happens
already early in the morning, clearly before the average event start-time. The possible source of iodic acid was





discussed earlier in chapter 3.2 and we hypothesize that the source of iodine at SMEAR I could be both; i) the
long distance transport from the Arctic Ocean combined to ii) the local emissions from the snow pack and
vegetation. The hypothesis of vegetation emitted iodine species is supported by the minor difference between
NPF (mostly marine) and non-event day (mostly continental) concentrations. At SMEAR I, HOMs are the
only species that are at a (marginally) lower level during non-event than NPF days indicating that the total
HOM concentration does not determine when NPF events occur. However, this does not exclude the possible
participation of certain HOMs in NPF together with sulfur compounds (Lehtipalo et al., 2018) or at later stages
of the NPF process, especially during particle growth. However, pure biogenic nucleation involving ions and
HOMs (Kirkby et al., 2016) seems not to be a major NPF pathway in Värriö.
Our measurements do not unveil the detailed mechanism of nucleation or growth of particles. We lack
measurements of ambient bases that are needed to stabilize sulfuric acid clusters in ambient conditions (e.g.
Almeida et al., 2013; Jen et al., 2014; Kirkby et al., 2011; Kürten et al., 2014; Myllys et al., 2018). With the
given observations comparing NPF days with non-event days it is likely that most regional NPF events require
sulfuric acid, but the NPF process can involve other compounds as well, especially IA and MSA, which show
higher concentrations on NPF days, very similarly that the results reported from Ny-Ålesund (Beck et al.,
421 2021).

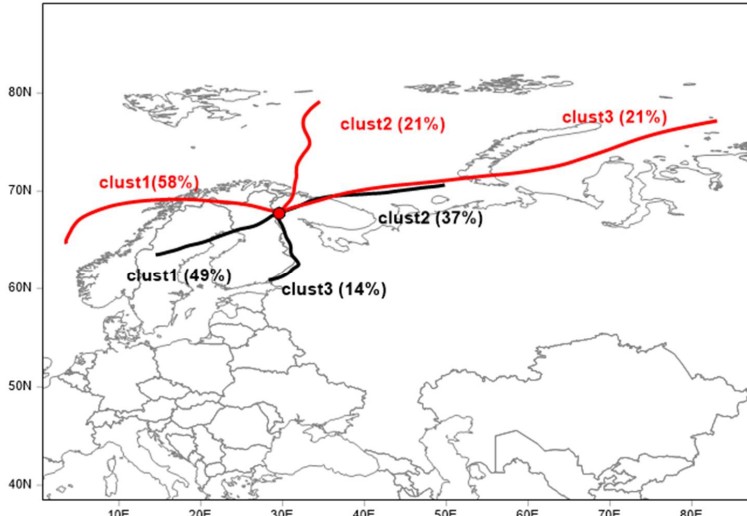


**Figure 10**. Trajectory cluster analysis of 72-hour back trajectories simulated at arrival height of 250 m a.g.l
and the NCEP/NCAR reanalysis data used as meteorological input. Red = NPF event, black = non-NPF
**Conclusions:**
We report ~7 months of nitrate based CI-APi-TOF measurements of sulfuric acid, methane sulfonic acid, iodic
acid and highly oxygenated organic compounds from a remote sub-Arctic field station SMEAR I in Finland.
The measurements aim to increase our understanding of the Arctic aerosol forming precursors and atmospheric
chemistry in more detail. The reason for measuring these compounds ~150 km north of the Arctic Circle is
simple; the Arctic is warming twice the speed as the planet on average. Lapland is already facing environmental
changes when e.g. woody plants disperse further north and influence the tundra ecosystem (Aakala et al., 2014;
Kemppinen et al., 2021). These changes will in turn affect the emissions of aerosol precursor gases, which
may have feedback effects on to the climate (e.g. Kulmala et al., 2020; Paasonen et al., 2013).



The area surrounding SMEAR I station has snow cover for almost 8 months a year. Accumulating snow during
the autumn is a good reservoir to e.g. halogens, similarly than in the high Arctic (and Arctic Ocean)
environment. The snow pack also acts as a cover for biogenic emissions entering the atmosphere from the
ground. Any changes in the temperature and snow cover in the sub-Arctic regions will effect on atmospheric
chemistry and composition that are undeniably changing the way aerosol particles form and what their number
concentration is in the region.
In this study, we report seasonal and monthly variations of SA, MSA, IA and HOM concentrations and find
all these compounds abundant in springtime. SA has a peak concentration in the spring, decreasing for the rest
of the seasons. We detect high concentrations of MSA and IA that are usually connected to marine and coastal
environments, although Värriö is located ~130 km from the nearest coast of the White Sea. While MSA is
abundant in the spring, summer and decreases to limit of detection levels for autumn, IA continues at the same
concentration throughout the seasons. It seems likely that these two compounds are connected to emissions
from phytoplankton or the Arctic ice pack and arrive to SMEAR I by long transport routes. In the case of iodic
acid, we suggest that the source of iodine emissions is a combination of transport and local emission from the
continental snow pack and vegetation at the site. Further work is needed to confirm this hypothesis.
The most striking correlation we found in HOM concentrations and ambient air temperature. The vegetation
at SMEAR I is the source of VOCs even in the snow covered spring season and these volatile gases are oxidized
into HOMs with different reaction rates depending on the oxidant. In the case of such strong temperature
controlled HOM concentrations, we conclude that HOMs in the mass range of 300 – 600 Th are most likely
products of monoterpene oxidation.
We also studied the abundance of these aerosol precursors separately during NPF and non-event days. We
observed that new particles at SMEAR I preferably form in relatively low temperatures (< 10°C), low relative
humidity that decreases with rising temperature during the day, ~10 ppbv higher ozone concentration than
during non-event days, high SA concentration in the morning and high MSA concentrations in the afternoon.
Cluster analysis of air masses show that NPF usually happens in marine air masses travelling to the site from
North west - West. All together, these are the first long term measurements of aerosol forming precursor from
the sub-arctic region helping us to understand atmospheric chemical processes and aerosol formation in the
rapidly changing Arctic.
**Data availability:**
All meteorological parameters, trace gas concentrations and aerosol data we downloaded directly from
smartSMEAR open access database ([https://smear.avaa.csc.fi/](https://smear.avaa.csc.fi/)). All mass spectrometric data are available on
request from the corresponding author.
**Author contribution:**
TJ, MS, TP and MK designed the experiments at SMEAR III and MS, NS, KN and TL carried them out. IY
made the NPF event analysis and RT calculated the back trajectories. TJ and KL wrote the manuscript with
contributions from all co-authors.
**Competing interests:**
Markku Kulmala is editor of ACP. Tuukka Petäjä is editor of ACP.
**Acknowledgements:**
We would like to thank the technical staff in Kumpula and Värriö, who keep the long-term measurements
going and helped with data collection, instrument calibrations, logistics and in data quality control and
assurance during the year. We acknowledge the important role our collaborators have in scientific discussion
and a special thanks goes to Alfonso Saiz-Lopez for iodic acid related discussion that helped to draft this





article. We thank the ACTRIS CiGas-UHEL unit for mass spectrometer calibration support and the tofTools
team for data analysis software.
**Financial support:**
The Academy of Finland via Center of Excellence in Atmospheric Sciences (project no. 272041) and European
Research Council via ATM-GTP 266 (742206), GASPARCON (714621) and Flagship funding (grant no.
337549) funded this work. We also received funding from the Academy of Finland (project no. 1235656,
296628, 316114, 315203, 307537, 325647, 33397, 334792 and 334514) "Quantifying carbon sink,
CarbonSink+ and their interaction with air quality" and Academy professorship (grant no. 302958). This work
was further supported by the European Commission via via project iCUPE (Integrative and Comprehensive
Understanding on Polar Environments, No 689443), the EMME-CARE project which received funding from
the European Union's Horizon 2020 Research and Innovation Programme, under grant agreement no. 856612,
Regional Council of Lapland (Värriön tutkimusaseman huipputkimus hyödyntämään Itä-Lapin
elinkeinoelämää, VÄRI, A74190) and Aatos Erkko Foundation.

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
