# Peer review of "Measurement report: Long-term measurements of aerosol precursor concentrations in the Finnish sub- Arctic boreal forest"

_Atmospheric Chemistry and Physics, 2021_

## Author Comment (AC1)

This manuscript presents 7 months of observations of aerosol precursor gases at the Finnish sub-Arctic field station SMEAR I. The authors combine their ambient pressure nitrate chemical ionization mass spectrometry measurements of sulfuric acid, methanesulfonic acid, iodic acid and the sum of highly oxygenated molecules with publicly available meteorological and aerosol data from the SMEAR I station. As the authors note, very few measurements of this nature and over such a long timescale are available to gain information on seasonal cycles of aerosol precursors in near-polar regions. For this reason, the present data set is quite interesting and the general analysis provide some interesting insights into aerosol formation chemistry. My main concern is that this analysis is somewhat cursory, and stops short of delving into detailed mechanistic analysis that such a data set can provide. However, I believe that this manuscript is deserving of publication in ACP given that the following mostly minor comments can be addressed.

We kindly thank you for your time to review our manuscript and we give you our detailed reply in the following sections in blue text.

Minor Comments:

(1) L104-105: what is meant by this statement? perhaps something is missing from the latter part of this sentence?

What we meant to say is that the chemistry behind NPF is never simple, even in pristine air. The sentences have been corrected now to clarify the meaning and says: However, the chemistry behind NPF is not simple, even in the pristine Arctic air.

(2) L143: what is the magnitude of the loss parameter in the 1 m stainless steel inlet line?

We completed the calibrations with the inlet tube so for this particular study we did not directly measure the loss parameter. From previous calibrations 1 m inlet line causes ~50% losses to sulphuric acid signals.

(3) L146-147 & L159-160: is the -50%/+100% uncertainty quoted only for sulfuric acid? what is the approximate uncertainty for non-sulfuric acid compounds for which the sulfuric acid sensitivity has been applied?

The uncertainty is given to represent all compounds mentioned in this study. HOM charging has been theoretically studied and it has been found that HOM charges in the nitrate CI-APi-TOF at kinetic frequency with $NO_3^-$ (Ehn et al., 2014), similarly to SA (Viggiano et al., 1982). In the case of iodic acid, the proton affinity of $IO_3^-$ is significantly lower than the charger ions and thus collision limit charging is again assumed (Sipilä et al., 2016). MSA has been observed to have the same calibration factor as SA in the nitrate-CIMS (Berresheim et al., 2002). If IA, MSA and HOMs would not charge at the kinetic frequency or close to it, it would lead to underestimation of concentrations in our manuscript. Thus the values reported in here are the lower limit estimations. This has now been clarified in the text.

L144-149 added a sentence:

If MSA, IA or HOMs do not ionize at the kinetic limit these concentrations could be underestimated and thus, the concentrations reported in here should be taken as lower limit values.

(4) L231: can the authors be more specific about what is meant by 'saturate'?

By saturation we mean that the concentration rises up very quickly and then stays stable for the rest of the day, almost like a flat line (= saturation), and then drops back in the evening. I used the word saturation because increasing concentration of a compound doesn't always lead to increasing signal in

your instruments but the detectors saturate and give a flat line even when increasing concentrations of the compounds. In this light the wording seems incorrect and I changed "saturation" to steady state.

(5) L290-304: as the authors note in the latter part of this paragraph, does it make sense to speculate about long range transport of HIO3? What is the approximate lifetime of $HIO_3$? Presumably its lifetime is short, and long range transport appears a very unlikely possibility.

You are correct, the lifetimes of iodine species are short, in this case they are reciprocal of the condensation sink (tau = 1 / CS). Since IA is only a bit smaller molecule than SA, the same can be estimated for IA. In Värriö, where CS is around 0.001 s$^{-1}$ (NPF days), the lifetime of IA should be around 15 minutes.

We did not speculate the long-range transport of IA, but its precursors, but the reviewer is in fact correct that the text isn't well written to explain this in detail. The precursor of IA — e.g., $CH_3I$ can be transported from the Arctic. This species has a lifetime of around 1 week and has both marine sources and terrestrial sources (Bell et al., 2002).

The text has been modified between L300-304 to:

"…suggests fast on-site chemistry, which is not consistent with long-range transport of iodic acid, but its precursor such as CH3I (Bell et al., 2002). Also, iodic acid life time against condensational loss is expected to be short, in the range of ~15 minutes, this suggests that HIO3 is formed close to or at the site of measurements."

Bell, N., Hsu, L., Jacob, D. J., Schultz, M. G., Blake, D. R., Butler, J. H., King, D. B., Lobert, J. M., and Maier-Reimer, E., Methyl iodide: Atmospheric budget and use as a tracer of marine convection in global models, J. Geophys. Res., 107( D17), 4340, doi:10.1029/2001JD001151, 2002.

(6) L305-313: why focus only on the melting of snowpack? Isn't photochemistry in the snow pack a more likely source? (e.g., Raso 2017, as cited by the authors)

The photochemistry in the snowpack was mentioned right after the melting snowpack: "This is possible due to the deposition of sea salts on snow particularly during dark periods that activate during the spring and are re-emitted to the atmosphere through heterogeneous photochemistry of iodide, and iodate ions" but we modified the "melting snowpack" to just "snowpack" for clarity.

(7) Figure 5: if the relationship between HOM concentration and temperature is most significant here, why not plot temperature on the x-axis and color by radiation?

We did plot the requested figure, but since the solar radiation is mostly <400 W m$^{-2}$ it doesn't really show the effect of solar radiation. However, it does nicely emphasize the correlation between HOMs and temperature and this we added it in the manuscript as panel B) next to the original picture.

[Figure]

(8) Figure 7: a diurnal profile of the condensation sink calculated from DMPS measurements would also be informative here, rather than only plotting total aerosol number concentration. (i.e., are the polar marine air masses also cleaner in terms of existing particles before NPF events occur, or does the very large concentration of aerosol precursors overwhelm any differences in condensation sink between event and non-event days?)

With the total number concentration, we want to show that NPF is producing a significant number of particles at the site. However, condensation sink is also an important parameter to be considered when trying to understand NPF and thus, we calculated the condensation sink from the DMPS data and added the figure in the manuscript:

[Figure]

The figure shows that the condensation sink is about 30% higher during non-event days than the event days.

(9) Figure 8/10: why do the authors think the $O_3$ is higher when air masses arrive from more northern polar marine regions?

Since ozone is photochemically produced in the presence of NOx (and VOC), we think that it relates more to the abundant solar radiation and availability of these trace gases than the airmasses arriving from North.

(10) Figures in general: many of the figures have very small axes labels, and are at a resolution that is difficult to read, please enlarge axes labels and legends and improve the figure resolution in the final version.

Thanks for the feedback, we increased the font size from 14 → 16 in all figures. Hopefully that is sufficient.

---

## Author Comment (AC2)

This measurement report by Jokinen et al. quantifies the seasonal and monthly variation of the important aerosol precursor gases sulfuric acid (SA), methane sulfonic acid (MSA), iodic acid (IA), and highly-oxygenated organic molecules (HOMs) from the remote field station SMEAR I in Värriö Northeastern Finland. The chemical species are measured almost constantly by a state-of-the-art nitrate chemical ionization atmospheric pressure interface time-of-flight mass spectrometer (CI-APi-TOF-MS) in the period between April-October 2019. In addition, the authors link the four chemical species and their seasonal concentrations, as well as meteorological parameters to new-particle formation (NPF) events classified by a Differential Mobility Particle Sizer (DMPS) and a Condensation Particle Counter (CPC).

The overall measurement reported is of good quality and its presentation is well structured, while the language is for the most part precise and fluent. As the authors mentioned, these are the first long-term measurements in a sub-Arctic region that can help the scientific community to understand regional chemical processes and formation of atmospheric aerosol as well as their dynamics in this remote region that is very sensitive to the Earth's rapid climate change. The scientific methods and assumptions are valid, and the interpretation of the measured data is discussed in detail, while the conclusions are limited and not always entirely clear. The conclusions therefore rather give a rough overview of the detailed mechanism of new-particle formation in sub-Arctic areas. As a measurement report, however, this work provides some interesting insights and serves as the basis for future and expanded research in this region. Since there are no major scientific concerns, I believe that this manuscript can be published in ACP as a measurement report after addressing the following mostly minor comments.

We kindly thank you for your time to review our manuscript and giving valuable comments how to improve it. We give you our detailed reply in the following sections in blue text.

*Minor comments:*

1. line159-61: Does this LOD of $4 \times 10^4 / cm^3$ also applies to other species than SA?

   Yes, it does, the instrumental background is used to count the LOD and it is at the same level for all the individual compounds in this study.

2. line197-99: Does these heatwaves have a substantial impact on the NPF at this site? Can the authors elaborate a little more on this in the discussion part as this may more often in the future?

   The increasing temperature will increase VOC emission and collaterally their oxidation to HOM will likely be emphasized with the current level of oxidants. However, heath wave conditions are (likely) not favourable conditions to NPF since condensation of low-volatility gases is favoured in colder temperatures (via the vapor pressure decrease with lower temperatures). Thus, heat waves could possibly inhibit NPF at the current precursor concentrations. However, if the VOC emissions leading to HOM concentrations increase, the growth of particles might be enhanced. The question remains whether the initial clusters are stable enough to reach that diameter where HOMs start to grow them even further or if pure biogenic nucleation (formation of dimers seems more plausible in hot conditions) could take a role in NPF during the heat wave periods in the future. Interestingly, the new figure plotted for Ref1 shows that the HOM concentrations seem to level above ca. 18C.

   We added the following sentences into the discussion (chapter starting from line 197):

However, heath wave conditions are likely not favorable conditions to NPF since condensation of low-volatility gases is favored in colder temperatures (via the vapor pressure decrease due to lower temperatures), but they may affect the oxidation chemistry of VOCs by promoting dimer formation.

We also added a figure of the temperature depence of HOMs in the manuscript and added some words (underlined) to a sentence (line 340): "It is striking how well the concentration of HOMs follow the air temperature (Figure 5) but seem to level above circa 18°C."

3. Figure 2: Since anthropogenic pollution plumes from the nearby smelters also affect the SMEAR I site, mixing ratios of $SO_2$ and $NO_x$ should be added here, if available, like it is done in Sipilä et al. (2021). This can help to understand various aspects of the photochemistry at this location (e.g., types of radicals and quantitative estimation, such as OH, $NO_3$, $RO_2$) and to separate NPF events from biogenic and anthropogenic sources.

We modified figure 2 to include more trace gas data, including $NO_x$, $SO_2$ and ozone. We decided to depict NPF days in two of the subplots to guide the eye, plotting NPF events on every panel made the figures look too crowded. We also added NPF dates in Figure 1 panel D. We hope that is sufficient for the reviewer.

4. line239-41: Why does the missing March data make the comparison (*more*) uncertain and in what perspective/direction. Can the authors elaborate on how far the diurnal cycle (e.g., peak concentration) or the overall SA concentration might change?

The missing data from spring (1 March – 3 April) equals ~30 % of the data allocated to spring data.

We added the following in the text:

"The SMEAR II data set that includes March data cannot be expected to be perfectly comparable with our data. However, as reported by Sipilä et al., 2021 the March data from the following year seems very similar concentration levels what we report in here for spring (max. ~2e6 $cm^{-3}$ and daily averages peak around 0.5e6 $cm^{-3}$). We expect that the SA concentrations are only marginally affected by the lack of March data, but that the level of HOMs or MSA or IA could be affected more due to very different meteorological conditions between the stations in springtime (SMEAR II is ~700 km South from SMEAR I). "

5. line243-53: Is there $NO+NO_2$ mixing ratios available during this period to state whether the HOMs at SMEAR I are mainly non-nitrate HOMs or organonitrates? This difference can also have a strong impact on HOM dimer formation and thus on NPF at this site. Taking the sum of HOMs (non-nitrate and organonitrates) from Sulo et al. (2021) at SMEAR II would result in a comparable diurnal cycle of total HOMs to SMEAR I. Sulo et al. (2021) shows a clear peak of organonitrates in summer at midday around $1x10^7$ mol./$cm^3$ while the non-nitrate HOMs are at ~ $0.4x10^7$ mol./$cm^3$. In the evening non-nitrate HOMs increase to about $0.8x10^7$ mol./$cm^3$ while the organonitrates begin to decrease but remain almost at the same concentration as the non-nitrated HOMs. It would be easier to follow the discussion and improve the reader's understanding of the various chemical species to include this section on HOMs in the later section starting at line 314.

The CI-APi-TOF data can be used to separate between nitrogen containing and other HOMs without the NO and $NO_2$ data (which are available from smartSmear). However, the scope

of this manuscript is to give an overview on the data, not to go into details of HOM chemistry. We acknowledge the fact that the data set could have been used to interpret HOM chemistry and its connection to NPF events in more details, but we decided that those will be discussed further in another separate manuscript in the future.

6.  Figure 3: It might be useful to add the diurnal cycle of the global radiation from the different seasons within the plots to link the aerosol precursor gases production to the ongoing photochemistry at SMEAR I station.

    The global radiation is now depicted in panel 3E with the following addition in the caption; Panel E depicts the seasonal variation of global radiation. The small (false) offset (6-7 W m$^{-2}$) in summer data is due to 24 h sunlight at Värriö.

[Figure]

7.  line314-16: Is the sum of HOM concentration in Yan et al. (2016) similarly calculated to this study (summation between 300-600 Th) and corrected for transmission to be comparable? Please add Yan et al. (2016) to the reference list.

    In Yan et al., 2016, they used the mass range of 201–650 Th as the total HOM concentration. We choose to use 300 Th as the lowest value due to peaks between 200-300 Th including e.g. one charger ion peak (($HNO_3$)$_3NO_3^-$ at 251 Th), iodic acid cluster ($HNO_3IO_3^-$ at 283 Th) and a persistent peak at 201 Th that is identified in Yan et al., 2016 in the transport factor. Those peaks would have distorted the HOM sum data reported in the manuscript.

    We do recognize that this sum of HOMs is not an ideal way to represent the complex HOM data set. However, this manuscript is meant to give the reader an overview of the data and to depict the abundance of HOMs (as group of compounds) at SMEAR I.

    Figure 5: I would also recommend plotting HOM concentration on the x-axis and temperature on the y-axis while coloring the global radiation. This might also reveal whether the behavior is linear or else. Does one data point represent a daily mean?

    We added the requested plot in the manuscript (see answers to Reviwe #1 also). One data point represents 30-minute average, in total this figure has 4960 data points.

8.  Figure 6+7: Since there is a clear seasonal (and even monthly) variation in aerosol precursor gases concentration, I wonder if this also leads to a diurnal variation of NPF events.

Therefore, a more distinct seasonal (or monthly) profile of median number size distribution and total particle concentration of the NPF events here would be more meaningful in my opinion.

Panels B and D in figure 2 are now updated to have NPF events that to cover the whole measurement period. It can be seen that spring is most abundant time for NPF to occur and that there is a long period during late summer (from mid-July to early-Sep) where only one NPF is observed. The whole period has only 33 events all together (2 type 1A-event,11 type-1B events and rest were type 2 events). Thus, the monthly or even seasonal would have ended up having just a few events and we thought it would be best to group them all together.

The seasonal variation on Ntot is plotted below and we added it with Figure 3 as panel F), hoping that this is satisfactory for the reviewer. The plot is showing the lower concentration in autumn and the impact of NPF in the spring as increase in Ntot in the afternoon and evening. We added a sentence to chapter 3.2. (5$^{th}$ row) to explain the figure shortly:

"We detect an increase in total aerosol number concentration on the spring evenings that is likely due to more frequent NPF events taking place at SMEAR I".

[Figure]

9. line364-71: The authors may add distinct numbers to the temperature, RH, and global radiation differences between event- and non-event days, as was done for the ozone concentration.

Text was edited as following;

"In Värriö, NPF events preferably happen in relatively low temperatures (1 – 8 °C) with a fast temperature rise in the early morning hours, lower and decreasing RH, dropping from 90% to ~55 %, during the NPF days compared to non-event days. NPF days have clearly higher global irradiance values (~450 m$^{-2}$ vs. ~200 m$^{-2}$) and about 10 ppbv higher ozone concentrations than non-event days."

10. line384-87: To what extent are the boundary layer dynamics affected during the day; especially concerning sunrise and sunset during the different seasons. What influence does it have on the measured precursor gas concentrations?

For an accurate assessment of the contribution of secondary aerosol formation to larger aerosol particles, the meteorological situation, including boundary layer dynamics, wet

deposition of particles, etc. should be considered. Sipilä et al., 2021 (ACP) reports that the SO2 concentrations are likely be much larger due to smaller PBL in winter time and thus, less mixed in the PBL. It is difficult to estimate the PBL effect on NPF without measurements at the site, thus we consider this something to develop in the future at the SMEAR I.

11. line391: Since the concentration of the measured gases depends on the wind direction and the air masses exposed to sources of volatile precursors, it would be great to add the wind direction to Figure 1.

We also thought of that and the figure we plotted is attached in here. It did not seem to give any essential information that we wanted to focus on in this article. We also thought to plot windroses (attached) but in the end, we decided go for the cluster analysis of air masses (Fig. 11) as we feel is the best way to see where the airmasses originated during the events vs. non-event days.

[Figure]

Fig i) SMEAR I meteorological parameters, RH % (top), wind direction (middle) and speed (bottom).

[Figure]

Fig ii) Windroses, NPF events on the left and non-events on the right.

12. line394-401: It looks like the air masses of cluster3 of event days and cluster2 of non-event days passed over the highly polluted region of Kola Peninsula. A representation of the

average particle number size distribution as a function of the different air mass clusters could therefore be very helpful here. In addition, as noted by Reviewer 1, the values of the condensation sink and the coagulation sink of small particles are important parameters for the classification of NPF events that should be added to the manuscript.

As reviewer no 1 also suggested, we added condensation sink to the manuscript. It can now be found in Figure 2H for the whole measurement period. The figure can be found below also.

The note of the air mass clusters we answer that many previous studies already report that Kola emissions contribute to elevated SO2 mixing ratios, particle formation events and particle numbers at the site (eg. Sipilä et al., 2021, Kyrö et al., 2014, Vehkamäki et al., 2004). Since the focus of this article is in the overview of aerosol precursor vapours, we did not see it necessary to plot the particle number size concentrations as a function of the clusters.

As suggested by both reviewers, we added the condensation sink in figure 2 from the whole measurement period, the figure is attached below.

13. Chapter 3.3: It would be helpful to have an overview showing the aerosol size distribution, wind direction, concentration of small particles or 2-7 nm negative ions, the particle sinks, and the NPF events over the entire measurement period, similar to Figure 2 in the accompanying article by Sipilä et al. (2021). In addition, the NPF events can be highlighted in Figure 1 and Figure 2 using background shading.

Thanks for the suggestion, you are correct. We added more descriptive plots covering the whole measurement period in Figure 2 (addressed earlier) and here (below) is the figure we added to describe the aerosol population and condensation sink for your convenience. The parameters suggested are indeed very necessary for the manuscript. The NPF events are depicted in the DMPS plot in red dots, just to avoid covering the particle size distribution data.

[Figure]

14. line409-414: Instead of using the sum of the HOMs, it may be useful to focus only on the ULVOC and ELVOC parts when parameterizing NPF, if possible. As mentioned by the

authors, however, HOM may play a minor role in nucleation, while it can strongly influence particle growth and compete for coagulation losses.

The HOMs are a fascinating subject to research, but as the purpose of this paper is to give an overview of SMEAR I CI-APi-TOF measurements, we do not see this approach necessary here. Thus, the volatility of HOMs is outside the scope of this article. We will definitely look into the HOM composition, its volatility and role in nucleation and growth in a separate manuscript in the future.

15. line415-417: Is there no evidence of mixed clusters between sulfuric acid, iodic acid, MSA, HOM, or possibly bases such as ammonia or amines observed by the nitrate CI-APi-TOF as reported by Sipilä et al. (2021) with the APi-TOF?

The CI-APi-TOF instrumental background (LOD $>10^4$ cm$^{-3}$) is much higher than corresponding ion APi-TOF background. We would **assume** that the level of possible SA clusters is very low ($<10^4$ cm$^{-3}$) like at SMEAR II (Jokinen et al., 2012) and thus not detected at SMEAR I. Since the detailed NPF mechanism is out of this manuscript's scope, we will leave this as an open question to solve.

16. To maximize the information provided in the abstract, please include unambiguous numbers of aerosol precursor gas concentrations and measured values rather than relative descriptions (e.g., "MSA shows a more distinct seasonal cycle with concentrations peaking in the summer of about $1 \cdot 10^8$ mol./cm$^3$"; or: "under relatively low temperature (1-8 °C)"). Why is the significant higher global radiation on NPF-event days not mentioned in the abstract?

Thank you for the suggestion, we added measured values into the abstract to make it clearer. It now reads:

"New particle formation events at SMEAR I happen under relatively low temperatures (1 – 8 °C) with a fast temperature rise in the early morning hours, lower (55% vs. 80% minimum value) and decreasing RH during the NPF days compared to non-event days. NPF days have clearly higher global irradiance values (~450 m$^{-2}$ vs. ~200 m$^{-2}$) and about 10 ppbv higher ozone concentrations than non-event days."

*Technical corrections:*

1. Please add the numbering of the individual chapters according to the text. Done.
2. Some passages require language editing. We did our best to edit the language.
3. line170: […] were downloaded […]? Fixed to "were".
4. line247: missing space between 'are one'. Fixed.
5. line298: missing 't' in events. Fixed.
6. line336-337: "[…] and detected aerosol precursor gases during NPF days […]." Fixed.
7. Figure 6: In the figure caption there is a 'during' too much. Second "during" deleted.
8. line369: "The meteorological conditions that favor NPF are thus similar to those at the SMEAR II station in Hyytiälä, […]". Fixed.

9. line377: Where does the reference 'Kulmala et al. (2013)' belong to? Furthermore, in this line add a 'C' to figure reference (e.g., Figure 8C). Kulmala deleted and "C" added.

10. line467: SMEAR III? Typo fixed.

11. The short summary needs to be revised in the second sentence.

---

## Author Response (AR2)

Thank you, Paul, for your efforts to make our article better, we appreciate it. Answers to your comment below in blue.

- There are still some more English grammar issues (e.g. in punctuation, see there is usually no comma before "that" unless it introduces parenthetical information) and in referencing (e.g. not in a coherent style, see e.g. lines 88, 86, 194).

- Please take care that all acronyms are defined at first occurrence in the main text (e.g. SMEAR, CU-APi-TOF, HOM).

- You introduce acronyms for compounds like sulfuric acid (SA) or iodic acid (IA) but then you are not using them consistently. I would suggest to harmonize this within the text.

-We fixed the punctuation issues, the referencing and went through the text and we now use the corresponding acronyms except in figure captions so they are easier to read. (Fixed and checked the following: NPF, CCN, VOC, SMEAR, HOM, DMS, IA, SA, MSA and RH)

- Line 119: The reference of He et al. (2021) is not really correct here since you talk about locations in the Arctic, while this study is a lab-based study.

-He et al. (2021) is cited because of the data in the supplementary material (Fig S9 & Fig S10). It reports iodic acid concentrations from Ny Ålesund, Svalbard and Villum, Greenland among locations in the Antarctic. Thus, we would like to keep the citation.

- Line 199 and 201: This is kind of repetitive since this information should be stated the "data availability section".

-We deleted the sentences saying "All meteorological parameters, trace gas concentrations and aerosol data were downloaded directly from smartSMEAR open access database (https://smear.avaa.csc.fi/) and all mass spectrometric data are available on request."

- Figure 2: Please add more tick-labels to panel G (one is not sufficient).

-You're right. More ticks are applied now.

[Figure]

- Line 380 is a bit repetitive compared to line 367.

-True, we deleted the sentence end from line 380 (", due to ocean surface acting as a major source of atmospheric iodine (Carpenter et al., 2013)")

- Line 386: There is something wrong in this parenthesis. Please also refer to the manuscript preparation guideline on how to abbreviate sections, figures, etc within the text. See https://www.atmospheric-chemistry-and-physics.net/submission.html

-We changed the referencing style throughout the text according to the guidelines and this part now reads: (discussed in details in Sect 3.3. and Figure 10).

- Line 395: No comma before "at".

-Fixed.

- Line 410: "at SMEAR"

-Fixed.

- Figure 6: I wondered about the range of the colorbar, especially in panel B, where overall concentrations were much lower (see Fig. 7A).

-I have updated my Matlab to a much newer version during the last months and the color map is slightly different between figures 6 and 7. It does not change the data plotted in the figures, just the color is slightly different. We double checked all data and they are correct in each separate plot. Hope that is ok.

- Data availability: The data presented in measurement reports must be openly accessible in accordance with the EGU data policy. Stating that "spectrometric data are available on request" is not optimal. I would recommend that you upload your data to a public repository and add the reference (link/DOI) here. See https://www.atmospheric-chemistry-and-physics.net/policies/data_policy.html for more details.

Mass spectrometric data, event analysis, condensation sink and anion concentration data are now available on Zenodo, https://doi.org/10.5281/zenodo.5879549. This information is added to data availability section.

- In the reply letter you mentioned that you added "The SMEAR II data set that includes March data cannot be expected to be perfectly comparable with our data." but this sentence is not there. Please double-check.

Added the missing sentence in its place now.

- In Fig 3E: The off-set is misleading. I would suggest to fill the gap or use lines instead.

We made a new figure and harmonized the coloring with the other panels also.

[Figure]

- Figure 5B: In the reply you mention that this plot shows 4960 points, which made me think. There could be the risk that e.g. the red points at higher global radiation are just covered by the blue points. You could test to apply a mesh (or grid) on your data and show the median or mean value of the global radiation per grid cell.

The main goal of these figures was to connect higher temperature with higher HOM concentrations. That is presented well by both panels. We originally only wanted to use panel A, because it has the same data and to our opinion it shows the relationship between global radiation, temperature and HOMs better. We added panel B to further emphasize the correlation between temperature and HOMs.

As you can see from the figure above (fixed panel in Fig. 3E), the median global radiation is around 400 W m$^{-2}$ or below that during summer and spring. Thus, there are only a few points that are red or orange in this plot, most of them are shown in the plot where the temperature is at its highest. Thus, we wish to keep these panels unchanged.

Looking forward to your final version!

We have done our best hoping you will like the manuscript better now that it is revised!

Best wishes, Tuija & co-authors